

# Modulation of attention to pain by goal-directed action: a somatosensory evoked potentials approach

Eleana A. Pinto[1,2], Stefaan Van Damme[1], Diana M. Torta[3] and Ann Meulders[3,4]

[1] Department of Experimental-Clinical and Health Psychology, Faculty of Psychology and Educational Sciences, Ghent University, Ghent, Belgium
[2] Department of Behavioural and Cognitive Sciences, University of Luxembourg, Esch-sur-Alzette, Luxembourg
[3] Health Psychology, Faculty of Psychology and Educational Sciences, KU Leuven, Leuven, Belgium
[4] Experimental Health Psychology, Department of Clinical Psychological Science, Faculty of Psychology and Neuroscience, Maastricht University, Maastricht, Netherlands

Corresponding author
Eleana A. Pinto,
Eleana.Pinto@UGent.be

## ABSTRACT

**Background:** Attentional processes are modulated by current goal pursuit. While pursuing salient cognitive goals, individuals prioritize goal-related information and suppress goal-irrelevant ones. This occurs in the context of pain too, where nonpain cognitive goal pursuit was found to have inhibitory effects on pain-related attention. Crucially, how pursuing nonpain motor goals affects pain-related somatosensory attention is still unknown. The aim of this study was to investigate whether nonpain motor goal pursuit would attenuate pain-related somatosensory attention.

**Methods:** Healthy volunteers ($N$ = 45) performed a robotic arm conditioning task where movements were paired with conflicting (pain and reward), threatening (only pain) or neutral (no pain and no reward) outcomes. To increase the motivational value of pursuing the nonpain motor goal, in the conflicting condition participants could receive a reward for a good motor performance. To examine somatosensory attention during movement, somatosensory evoked potentials (SEPs; N120 and P200) were obtained in response to innocuous tactile stimuli administered on a pain-relevant or pain-irrelevant body location. We expected that the threat of pain would enhance somatosensory attention. Furthermore, we expected that the possibility of getting a reward would inhibit this effect, due to pain-reward interactions.

**Results:** Against our predictions, the amplitude of the N120 did not differ across movement types and locations. Furthermore, the P200 component showed significantly larger SEPs for conflicting and threat movements compared to neutral, suggesting that the threat of pain increased somatosensory attention. However, this effect was not modulated by nonpain motor goal pursuit, as reflected by the lack of modulation of the N120 and P200 in the conflicting condition as compared to the threat condition. This study corroborates the idea that pain-related somatosensory attention is enhanced by threat of pain, even when participants were motivated to move to obtain a reward.

## INTRODUCTION

Attention is the mechanism through which sensory inputs are selected to enter awareness. In daily life, we are often exposed to multiple environmental stimuli. In such a context, perceptual priority is defined by cognitive processes that guide attention towards goal-related information (top-down attention; *Corbetta & Shulman, 2002*; *Theeuwes, 2010*). Furthermore, attention can also be captured in a bottom-up fashion by physically salient stimuli (salience-based attentional capture), task-relevant sensory inputs that are congruent with the ongoing goal (contingent attentional capture; *Anderson & Folk, 2010*; *Folk, Remington & Johnston, 1992*) and by stimuli that are physically non-salient but that signal a reward (*Anderson, Laurent & Yantis, 2011*; *Della Libera & Chelazzi, 2006*, *2009*; *Laurent et al., 2015*; *Peck et al., 2009*; *Raymond & O'Brien, 2009*). These modes of attentional control work jointly to determine attentional prioritization (*Anderson, 2015*; *Corbetta & Shulman, 2002*; *Yantis, 1998*). Consequently, attention is deployed in accordance with the current goals towards behaviorally salient stimuli.

These modes of attention selection apply also to pain-related attention. In fact, pain can unintentionally capture attention (*Eccleston & Crombez, 1999*), interrupting the ongoing behavior and prioritizing appropriate actions to protect or escape from bodily threats (*Legrain et al., 2009*; *Robison et al., 2021*; *Van Damme et al., 2009*). This occurs because pain is a salient sensory input, thus its prioritization represents an adaptive mechanism that is functional to maintaining bodily integrity. However, pain-related attention is also susceptible to top-down control (*Legrain et al., 2009*; *Legrain, Crombez & Mouraux, 2011*; *Seminowicz & Davis, 2007*). In fact, allocating to or diverting intentionally attention away from pain can impact its experience (*Buhle & Wager, 2010*; *Crombez, Van Damme & Eccleston, 2005*; *Hoegh, Seminowicz & Graven-Nielsen, 2019*; *Notebaert et al., 2011*; *Rischer et al., 2020*; *Roa Romero et al., 2013*; *Van Damme et al., 2008*; *Verhoeven et al., 2010*, *2011*; *Vossen et al., 2018*). Similarly to attentional selection modes presented previously (*Corbetta & Shulman, 2002*; *Legrain et al., 2009*; *Yantis, 1998*), top-down selection of attention is a goal-directed act of prioritizing information that is relevant for the current action (*Legrain et al., 2009*).

Starting from this conceptualization, the motivational account of attention to pain suggests that pain-related attention should be framed within a context of goal pursuit (*Torta et al., 2017*; *Van Damme et al., 2010*). Within this framework, attention is selecting the information for actions needed to pursue goals, whereas goal-irrelevant information is inhibited. Hence, given that pain occurs in a context of multiple and often competing goals (for example, persisting in running despite experiencing muscle pain in order to complete a marathon), the ability of pain to capture attention depends on both the features of the pain and on those of the current goal. The motivational account of pain-related attention is supported by experimental studies that showed that engaging in a salient pain-unrelated cognitive task reduced pain-related attentional processes (*Buhle & Wager, 2010*; *Legrain, Crombez & Mouraux, 2011*; *Schrooten et al., 2012*; *Seminowicz & Davis,*

*2007*; *Van Damme et al., 2009*, *2010*, *2012*; *Verhoeven et al., 2010*), and that this effect is supposedly driven by inhibitory abilities (*Verhoeven et al., 2011*) and working memory (*Legrain, Crombez & Mouraux, 2011*). These results demonstrated that pursuing a valued cognitive goal may induce a top-down inhibition of attention to pain, and this mechanism is proposed to protect the goal-directed behavior from distracting goal-irrelevant information (*i.e.*, goal-shielding; *Legrain et al., 2012*, *2013*; *Van Damme et al., 2010*). However, the experimental studies supporting the motivational account of pain-related attention mostly focus on how pursuing a *cognitive* goal impacts pain-related attention, leaving unexplored the *motor* domain. Importantly, cognitive psychology models seem to suggest that action can modulate attentional capture (*Kerzel & Schönhammer, 2013*; *Kirsch, Kitzmann & Kunde, 2021*; *Welsh & Pratt, 2008*), by determining attentional deployment towards salient stimuli (*Castaneda & Gray, 2007*; *Hommel, 2010*; *Welsh, 2011*; *Wood et al., 2011*). However, to our knowledge, most of the studies that investigated the link between action and attention mostly focused on visual attention (*Kirsch, Kitzmann & Kunde, 2021*) leaving unexplored the somatosensory domain. The only studies that looked at somatosensory processing during motor action found that anticipating (*Clauwaert et al., 2018*) and executing (*Clauwaert et al., 2020*) motor actions conditioned with painful outcomes, enhanced the processing (assessed using electroencephalography; EEG) of tactile inputs in a pain-relevant body location. Attentional processes presumably drive this effect, that is, individuals attend more toward the pain-relevant location when performing a movement associated with the threat of pain. However, the experimental paradigm implemented in these studies (*Clauwaert et al., 2018*, *2020*) consisted of simple reaching motor actions and left unexplored the potential modulatory role of pursuing nonpain goals while moving. Crucially, having to move despite pain in the moving body part is a very common experience that individuals might need to face when pursuing daily goals. However, no research investigated how executing goal-directed motor actions affects pain-related attention. Hence, this represents an important knowledge gap that we aimed to address with the current study. With this purpose, healthy participants performed a robotic arm conditioning task in which three motor actions were conditioned with threatening (pain), neutral (no consequences) or conflicting (pain and reward) outcomes. Somatosensory evoked potentials (SEPs, namely the N120 and the P200 components) were obtained during action execution in response to innocuous tactile stimuli administered to either a pain-relevant or a pain-irrelevant location.

We hypothesized to replicate that the threat of pain enhances pain-related somatosensory attention (*Clauwaert et al., 2018*, *2020*). This would be reflected by (1.a) a larger N120 in the pain-relevant location in the threat movements as compared to neutral movements and (1.b) a location unspecific enhancement of the P200 component for threat movements compared to neutral movements. Furthermore, we expected the nonpain motor goal pursuit to inhibit pain-related somatosensory attention as reflected by (2.a) a smaller N120 component in the pain-relevant location in the conflicting movements as compared to threat movements and (2.b) a location unspecific reduction of the P200 component for conflicting movements compared to threat movements. The hypotheses

and analysis plan were registered before the start of data collection on Open Science Framework (osf.io/vbgru).

# MATERIALS AND METHODS

## Participants

Forty-five healthy, pain-free volunteers (35 women and 10 men, mean age 22.47 years (range 18–44)) gave their written informed consent and participated in the experiment. Sample size calculation was performed with GPower software (G*power 3.1 (*Faul et al., 2007*) to obtain 95% power to detect medium effect size (based on Cohen's effect size convention $f = 0.25$; *Faul et al., 2007*)). Sample size calculation was also based on the effect sizes on previous studies with a similar paradigm in which medium-sized effects were observed (*Clauwaert et al., 2018*, *2020*). Students were recruited *via* the SONA system of Maastricht University, while members of the general population were reached *via* social media platforms, flyers and word-of-mouth. After participating in the study, of approximately 2 h, participants choose to receive either two course credits or a gift voucher worth 15 Euros.

The exclusion criteria included pregnancy, left-handedness, chronic pain anywhere in the body, acute pain anywhere between the right shoulder and right hand, uncorrected hearing and/or vision problems, current or history of cardiovascular disease, neurological disorder, current or history of a psychiatric disorder (*e.g.*, depression, panic/anxiety disorder), any other serious medical condition and an electronic implant (*e.g.*, cardiac pacemaker) or having been advised by a general practitioner to avoid stressful situations. Participants were informed about the potential discomfort of pain stimuli before signing the informed consent. The study was conducted in accordance with the principle of the Declaration of Helsinki and the Ethics Review Committee Psychology and Neuroscience (ERCPN) of Maastricht University (number ERCPN- 233_18_02_2021_A1). One participant was excluded from the analysis because of the bad quality of the EEG data that could not be pre-processed.

## Pain stimuli

A commercial constant-current square wave electrical pulse stimulator (DS7A) delivered pain stimuli, consisting of brief electrocutaneous stimuli (duration 2 ms). These stimuli were administered *via* two reusable stainless steel disk electrodes (8 mm diameter with 30 mm spacing; Digitimer, Welwyn Garden City, UK). The electrodes were filled with K–Y gel (Reckitt Benckiser, Slough, UK), and attached to the right wrist or triceps tendon (counterbalanced across participants) of the right (dominant) arm using a Velcro strap (*Meulders et al., 2016*). The stimulus intensity was determined for each participant individually using a standardized calibration procedure (see Pain Calibration under *Procedure*).

## Tactile stimuli

Two resonant-type tactile probes (C-2 TACTOR; Engineering Acoustics, Inc., Casselberry, FL, USA; *Van Hulle et al., 2013*) were used to administer vibrotactile stimuli. These stimuli

lasted 200 ms, their frequency was 300 Hz and their intensity 0.04 watts; these parameters were set *via* a self-developed software program. The probes were attached with double-sided tape rings to the skin of the participants in the pain-relevant location (proximal to electrodes used for the pain stimulation) and another one in a pain-irrelevant location (distal from the electrodes used for the pain stimulation, depending on the counterbalancing the dominant wrist or triceps tendon).

## Robotic arm task

A three-degrees of freedom robotic arm (The HapticMaster) was used to perform the motor task. This robotic arm allows for a range of movements restrained to 0.36 m depth, 0.40 m height and 1 m width (FCS Robotics; Moog Inc, East Aurora, NY, USA). The movements were confined to a two-dimensional, horizontal movement plane. As the participants moved the robotic arm, they could see their movements on the screen represented by a green ball. The task included different reaching movements passing through one of three arches positioned mid-way the movement plane. Movements were executed using the right hand by holding the sensor at the extremity of the robotic arm. Participants seated in front of an LCD screen, which was used to visualize the task set-up, the participants' movements, and the ratings scales for self-reports. A Windows 10 compatible triple foot switch (USB–3FS-2; Tokyo, Japan) was used to answer questions presented occasionally during the task. The left pedal was used to scroll to the left, the middle to scroll to the right on the rating scale, and the right foot pedal was used to confirm the participant's answer. While participants responded to these questions, the robotic arm was blocked.

A green ball indicated the position of the robotic arm sensor representing participants' own movement on-screen. During each trial, a movement initiation cue (coloring of the trajectory arch in yellow) was presented for 1,000 ms. This cue instructed participants through which of the three arches they were required to move. Participants were told to start moving as soon as the trajectory arch turned black again (movement execution cue). A red ball was shown as a tracing cue, meaning that it moved ahead of the green ball and participants received the instruction to accurately follow the trajectory of the tracing cue. The three possible trajectories were equivalent in terms of force needed to execute the movement. The robotic arm only allowed movement during the movement execution window (during other phases it was haptically blocked). Although all arches were displayed on the screen, only the instructed movement was possible (*i.e.*, the movement through the arch that was colored in yellow during movement preparation phase). This meant that the movement of the robotic arm through those arches was not possible (if participants tried to go through the wrong arch the robotic arm would block). Boundaries were set along the trajectory and the robotic arm was haptically blocked beyond the boundaries so that participants had limited movement space around the defined trajectories. The boundaries were not visible on the movement plane. This set of boundaries determined the trajectory that the tracing cue would follow through each of the arches. This way, it was made sure participants stayed within the limit of the trajectory of the tracing cue. Each trial ended once the green ball reached the end of the movement

plane where a target arch was presented. The robotic arm was then automatically repositioned to the start location (in 750 ms). As the participants moved the robotic arm, aiming for the indicated arch, the other two non-available arches were colored grey (for an illustration of the trials structure see Fig. 1).

Each of the three possible movements was paired with a specific outcome. One movement was followed by a pain stimulus (threat movement), one movement was accompanied by pain and reward (conflicting movement, see *Reward manipulation*). In the neutral movement, there were no consequences. The order by which the movement trajectories were displayed on the screen was counterbalanced across participants so that the trajectories serving as threat, conflicting, and neutral were located on different positions to avoid confounding effects due to their location.

## Reward manipulation

To increase the motivational value of the movement, a reward manipulation was used. At the beginning of the experiment, participants were asked to choose a prize worth 100 Euros from a list. During the conflicting movement trials only, a lottery ticket image was presented on the right upper corner of the movement plane. Participants were told that the image of the lottery ticket indicated that their motor accuracy was being measured and that their motor accuracy at these trials would determine the number of lottery tickets that they could receive. This was done to increase the motivational value of performing the movement accurately. Moreover, at the end of each block, they received feedback regarding their motor accuracy together with the cumulative number of lottery tickets collected up to that point. However, although we did measure participants' motor accuracy, the feedback presented consisted of a bogus percentage accuracy meter (programmed to show a fixed value for all participants). This was done to ensure that all participants received the same reward. The sequence of lottery tickets gained was: 12 and 14 (during acquisition), 15, 16 and 18 (during test phase) and 20 after the free choice phase. An exemplary picture of the bogus percentage accuracy meter is shown in the last screen of Fig. 1.

## Procedure

The experiment started with the pain calibration procedure. Subsequently, the experiment consisted of four experimental phases, namely a practice, acquisition, test, and free choice phase.

## Pain calibration

For the calibration, a series of electrocutaneous stimuli of increasing intensity were manually administered following a standardized procedure based on previous studies (*Gatzounis & Meulders, 2020*; *Glogan et al., 2020*; *Meulders et al., 2016*). Hence, data were collected as described previously (*Vandael, 2022*). Specifically, after each stimulus participants rated how painful it was on a 0-to-10-point rating scale, where 0 is labelled as "I feel nothing"; 1 as "I feel something, but this is not unpleasant; it is only a sensation", 2 as "the stimulus is not yet painful, but is beginning to be unpleasant", 3 as "the stimulus

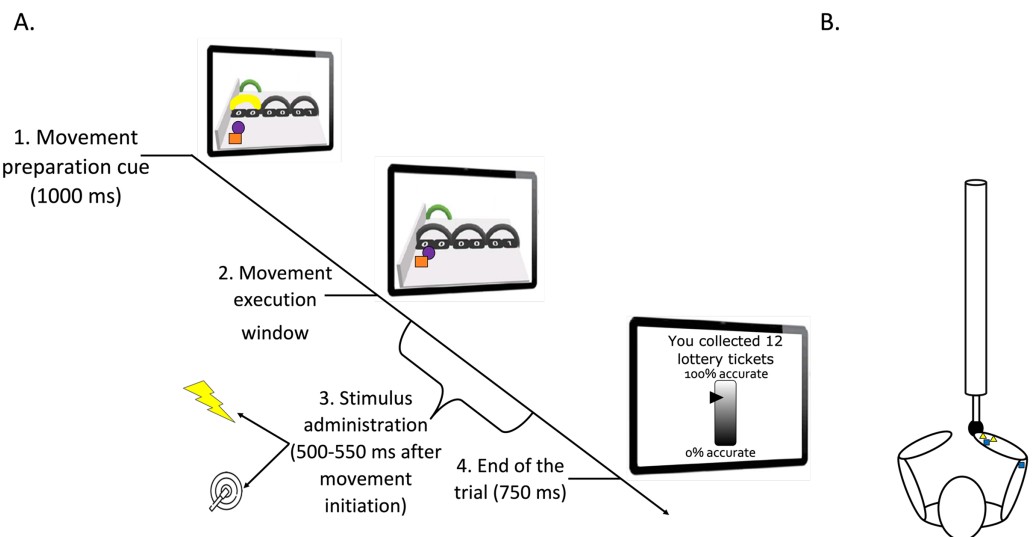

**Figure 1** **(A) Timeline of a trial (test phase). (B) Graphic representation of the stimuli positioning.** (A) Participants looked at the screen and prepared the movement related to the yellow arch (1. Movement preparation cue). They executed the movement after it was no longer colored (2. Movement execution window). During this latter phase, a pain (25%) or a tactile (75%) stimulus was administered between 500 and 550 ms after movement initiation (3. Stimulus administration). The trial ended 750 ms after completion of the movement (4. End of the trial). Then, the robotic arm automatically returned to the start position. A representation of the motor performance feedback is also shown on the last computer screen. Please note that for accessibility reasons, the tracing cue and the cue representing participants' own movement on-screen are here represented as a purple ball and orange square, respectively. However, in the actual paradigm, a red and a green ball were used (as described in the manuscript text). (B) Pain electrodes placed on the wrist (yellow triangles) and tactors in the pain-relevant (right wrist) and in the pain-irrelevant location (triceps tendon; dark blue squares). Note that in some participants the electrodes for the pain stimuli were located on the triceps tendon to counterbalance the stimulus location.

starts being painful"; and 10 as "this is the worst pain I can imagine". The participants were encouraged to select a stimulus that was "significantly painful and demanding some effort to tolerate", roughly representing a stimulus intensity of 7 or 8, on this calibration scale. Participants' physical intensity of the electrocutaneous stimulus selected with the calibration procedure was 31.44 mA (SD = 1.44) and the mean self-reported stimulus intensity was 7.67 (SD = 0.56).

## Practice phase

A practice phase was included to make sure that participants understood the task and knew how to answer the self-report questions by using the triple foot switch. For this purpose, each of the three movements (conflicting, threat, and neutral) was performed twice (six trials in total) and no pain or tactile stimuli were presented. After the practice phase, participants were trained on how to use the triple foot switch to answer the questions. The corresponding arch was colored green as soon as the question was presented on the screen to indicate to which trajectory the question pertained and remained colored until the question was answered. The following questions were presented on the screen above the movement plane after completion of the practice trials: *"To what*

**Table 1 Study design.**

| Movement type | Location | | |
| --- | --- | --- | --- |
| | Pain-relevant | Pain-irrelevant | Total |
| Threat | 48 | 48 | 96 |
| Neutral | 48 | 48 | 96 |
| Conflicting | 48 | 48 | 96 |
| Tot. | 144 | 144 | 288 |

Note:
Overview of the trials per condition. Note that within this study design, 75% of the trials were paired with a tactile stimulus, while the remaining 25% of the trials were paired with a pain stimulus.

extent would you expect a stimulus when moving through the green arch?" (Pain expectancy), "How afraid would you be to move through the green arch?" (Fear of movement) and "How painful was the electrical stimulus you received when you moved through the green arch?" (Pain intensity). Since the neutral movement was not paired with a pain stimulus, the latter question was not presented for this trajectory. Participants were asked to answer using the 0–100 Visual Analogue Scale (VAS; 0 = "not at all" and 100 = "very much") presented on the screen.

## Acquisition phase

This phase consisted of two blocks of 12 trials, with four trials per movement type per block (24 trials in total). During this phase, participants learned that each movement was paired with different outcome. The threat movement was paired with a pain stimulus in 75% of the trials. The conflicting movement was paired with pain in 75% of the trials, but also with a reward (See *Reward manipulation* for more details). The neutral trials were not paired with any consequences. For the threat and conflicting movements, a 75% reinforcement rate was used—instead of a 100%—since anticipating uncertain threat elicits stronger conditioning responses (*Bennett, Dickmann & Larson, 2018*; *Morriss, Zuj & Mertens, 2021*). At the end of each block, participants answered the three questions (pain-related fear, pain expectancy and pain intensity) for each trajectory (for the full text of these questions see *Practice* section).

## Test phase

The test phase consisted of three blocks of 96 trials (32 trials of each movement type per block), for a total of 288 trials. In addition, tactile stimuli were administered in 75% of the trials. Pain stimuli were administered in the remaining 25% of the trials during the threat and conflicting movement (this was done to counteract quick extinction; *Glogan et al., 2020*). In Table 1, we illustrate the study design and the number of trials per each condition. The stimuli (pain and tactile) were administered between 500 and 550 ms from the onset of the movement. Furthermore, the same questions as in the practice and acquisition phase were presented. Throughout the test phase, the EEG signal was recorded (see *EEG recording and analysis* for more details).
## Free choice

Finally, the last phase was a free choice phase in which participants could freely decide through which trajectory they could move. This block was included as a manipulation check to measure how motivated participants were to perform each movement. For 12 free choice trials of this phase, an auditory start cue (1,000 ms) replaced the visual movement initiation cue (*i.e.*, the yellow coloring of the arch). No arch color was changed, and the tracing cue was not displayed. Participants were simply asked, on each trial, to freely choose one arch to move through with the green ball. The trial ended once the movement was completed after which the robotic arm was repositioned to the start location and fixed until the start of the next trial. While no tactile stimuli were administered, pain stimuli were again administered in 25% of trials between 500 and 550 ms from movement initiation (only for threat and conflicting trials).

Two additional questions were asked assessing the participants' personal importance of reward and avoidance respectively with the questions: *"How important was it for you to gain more lottery tickets?"* (Reward importance) and *"How important was it for you to avoid the pain stimuli?"* (Avoidance importance) using a 0–100 Visual Analogue Scale (VAS; 0 = "not at all" and 100 = "very much").

## Outcomes

Our primary outcome, namely pain-related somatosensory attention, was operationalized as the mean amplitude of the N120 and P200 components evoked by the tactile stimuli at the pain-relevant and pain-irrelevant location. In addition, pain-related somatosensory attention was also assessed *via* the mean amplitude of the SEPs (N120 and P200) elicited by the pain stimuli (during the conflicting and threat trials only). The latter (pain-evoked) SEPs were analyzed as secondary outcomes. Note that the pain stimuli were implemented in this study for conditioning purposes, which is pairing a motor action with painful or conflicting outcomes (as reported in our pre-registration osf.io/vbgru). We also included manipulation check measures of pain intensity, fear of movement, pain expectancy (per movement type) and self-reported measures of questions asked at the end of the free choice phase (avoidance and reward importance). Moreover, we obtained measures of individual preferences during the free choice phase (number of trials participants selected each of the three trajectories when they could freely choose). Finally, we measured motor performance accuracy per movement type (operationalized as the number of times the boundaries were hit).

## EEG recording and analysis

EEG was recorded with a 32-channel portable system (EEGO Sports; ANT neuro system, Hengelo, Netherlands). The active electrodes were placed according to the international 10–20 setting system. The ground electrode was located in the active-shield cap fronto-centrally between the FPz and the Fz electrode. Impedances were kept <10 kΩ, using an electroconductive paste. The sampling rate was set at 2,000 Hz. No online filters were applied. Data were pre-processed off-line by using a Matlab open-source toolbox for EEG analysis: Letswave 7 (www.letswave.org/).

The continuous EEG data were re-referenced offline to an average of all electrodes. Data were filtered with a 4th order Butterworth 0.3–30 Hz band pass filter. Independent component analysis (ICA) was performed to correct artefacts (muscular, bad channel and ocular) and blinks. The EEG signal was epoched in segments from −200 to 500 ms relative to the onset of the tactile stimulation. An automatic artefact rejection was applied to these segments based on a minimum/maximum amplitude check (−75 µV and 75 µV respectively; *Clauwaert et al., 2020*). Next, a baseline correction (−200 ms, 0 from the onset of the stimulus) was carried out and epochs were averaged per each movement condition and stimulus location (the latter for the tactile stimuli only). As a result of the averaging procedure, we obtained the following waveforms (based on movement type and stimulus location): (a) Threat movement—Pain-irrelevant location, (b) Threat movement—Pain-relevant location, (c) Neutral movement—Pain-irrelevant location, (d) Neutral movement—Pain-relevant location, (e) Conflicting movement—Pain-irrelevant location, (f) Conflicting movement—Pain-relevant location. Additionally, we obtained waveforms elicited by the pain stimuli in the threat movement and in the conflicting movement (secondary outcome). Note that the neutral movement was never paired with a pain stimulus, therefore no SEPs were obtained for this movement type. Similarly, for the pain stimuli, the averaging procedure only resulted in SEPs from two conditions, namely conflicting and threat movements.

A collapsed localizer (grand-grand average across conditions) was used to identify the waveforms peaks and their scalp distribution (*Luck & Gaspelin, 2017*). With this procedure and consistently with previous research (*Clauwaert et al., 2018*, *2020*), the following peaks were identified: a negative peak around 130 ms (for the electrodes Fz, Fc1, Fc2 and Cz) and a positive one around 214 ms (for the electrodes Cz Fc1, Fc2, Cp2 and Cp1) from stimulus (tactile) onset. Similarly, for the pain stimuli, a negative peak was identified at 118 ms (for the electrodes Fz, Fc1, Fc2 and Cz) and a positive peak at 220 ms (for the electrodes Cz Fc1, Fc2, Cp2 and Cp1) from the onset of the pain stimulus. We selected 50 and 100 ms intervals around averaged waveform peaks of all participants and all conditions for the N120 and the P200 component respectively (*Clauwaert et al., 2018*, *2020*). With this purpose, we computed the waveform mean area (amplitude by time). Statistical analysis was performed on the pooled (averaged) channels where the peaks were identified.

## Data analysis

Statistical analyses were performed in JASP (2020 Version 0.13.1; *JASP Team, 2020*).

To test whether the threat of pain (hypothesis 1.a) or nonpain motor goal pursuit (hypothesis 2.a) modulates somatosensory attention (operationalized as the amplitude of the N120 component elicited by tactile stimuli applied to the pain-relevant location), we used a 2 (stimulus location: pain-relevant *vs* pain-irrelevant) × 3 (movement type: threat, neutral, conflicting) repeated measures analysis of variance (RM ANOVA). For significant main results or interactions, we used *post-hoc* tests. For multiple comparisons, we used a Bonferroni correction (*i.e.*, we divided the critical *p* value (α) by the number of comparisons) to test for significant differences in the SEPs amplitude between each pair of
conditions. Specifically, we compared threat and neutral trials for hypothesis 1.a and threat and conflicting trials for hypothesis 2.a.

Similarly, to test whether the threat of pain (hypothesis 1.b) or nonpain motor goal pursuit (hypothesis 2.b) modulates general non site specific somatosensory attentional processes (operationalized as the amplitude of the P200 component), we used a 2 (stimulus location: pain-relevant *vs* pain-irrelevant) × 3 (movement type: threat, neutral, conflicting) repeated measures analysis of variance (RM ANOVA). For significant main results or interactions, we used *post-hoc* tests (with Bonferroni correction for multiple comparisons) to test for significant differences in the SEPs amplitude between each pair of conditions. Specifically, we compared threat and neutral trials for hypothesis 1.b and threat and conflicting trials for hypothesis 2.b. Note that for this component, we expected a location unspecific effect.

As a secondary analysis, we explored whether the amplitude of the pain-evoked SEPs was decreased in the conflicting compared to the threat movement. For this purpose, we used a t-test for both the N120 and the P200. Cohen's d or partial eta squared is reported to calculate effect size (respectively, for the t-tests and ANOVAs). For the analyses in which Mauchly's test of sphericity indicated a violation of the assumption of sphericity, a Greenhouse-Geisser sphericity correction was applied to the degrees of freedom. To test whether the conditioning was successful, two different one-way ANOVAs (within factor: movement type) were performed on the mean score of the manipulation check items (*i.e.*, fear of movement, pain expectancy). Differences in pain intensity were checked using a t-test since there were only two movements paired with pain stimuli (no painful stimuli were administered during the neutral movement). To test for significant differences in the free choices a one-way ANOVA was used with the factor movement type. Furthermore, to test whether the type of movement affected the motor task performance a one-way ANOVA (within factor: movement type) was performed on the motor performance score.

## RESULTS

### EEG somatosensory evoked potentials

#### *Primary outcomes*

*Tactile N120.* The 2 × 3 RM ANOVA (stimulus location (pain-irrelevant *vs* pain-relevant) × movement type (threat, neutral and conflicting)) on the tactile N120 SEPs did not show significant effects of movement type, $F_{(1.514,43)} = 1.488$, $P = 0.23$, $\eta_p^2 = 0.03$, or stimulus location, $F_{(1,43)} = 0.002$, $P = 0.96$, $\eta_p^2 = 0.0005$, or interactions, $F_{(1.582,43)} = 0.289$, $P = 0.698$, $\eta_p^2 = 0.007$. Contrary to our hypothesis, threat of pain and nonpain motor goal pursuit (hypothesis 1.a and 2.a respectively) did not impact pain-related somatosensory attention, as the amplitude of this component was not influenced by the type of movement or by the stimulus location.

*Tactile P200.* The 2 × 3 RM ANOVA (stimulus location (pain-irrelevant *vs* pain-relevant) × movement type (threat, neutral and conflicting)) on the tactile P200 component showed a main effect of movement type, $F_{(1.73,43)} = 17.20$, $P = < 0.001$, $\eta_p^2 = 0.29$, but no main effect of stimulus location, $F_{(1,43)} = 1.41$, $P = 0.32$, $\eta_p^2 = 0.02$, nor

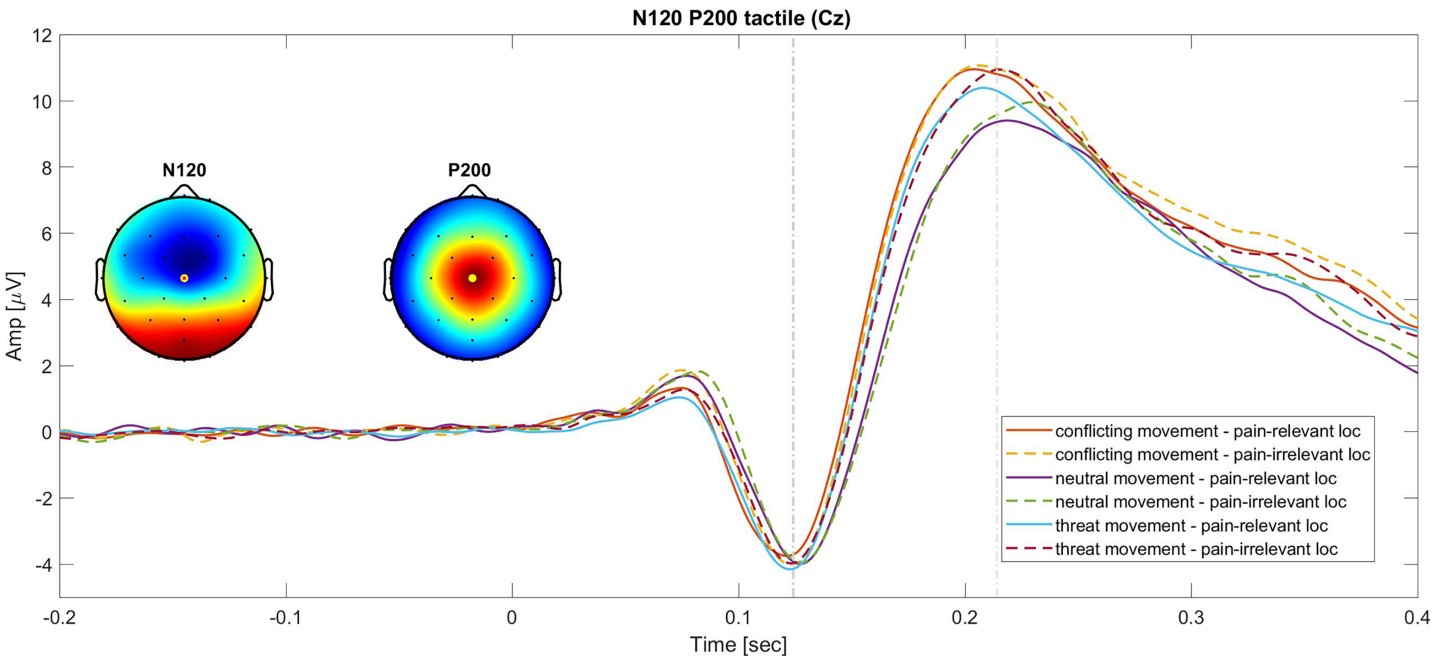

**Figure 2 Grand average of the N120 and P200 SEPs at the Cz electrode.** Continues lines show the SEPs elicited for different movement types in the pain-relevant location condition and dotted lines represent the SEPs elicited in the pain-irrelevant movement conditions.

interaction, $F_{(2,43)} = 0.46$, $P = 0.63$, $\eta^2_p = 0.01$. The *post-hoc* follow up analysis revealed that the P200 component was significantly larger when the tactile stimuli were administered during a conflicting movement (M = 6.71, SD = 4.08) compared to neutral (M = 5.55, SD = 3.23; $t_{(43)} = 5.75$, $P_{bonf} =< 0.001$, $d = 0.87$), but not compared to threat movements (M = 6.33, SD = 3.72; $t_{(43)} = 1.86$, $P_{bonf} = 0.20$, $d = 0.28$) and larger during the threat movement compared to neutral $t_{(43)} = 3.88$, $P_{bonf} < 0.001$, $d = 0.59$. As predicted (hypothesis 1.b), the threat of pain induced an unspecific increase of this component. This was reflected by a larger P200 in the threat movements as compared to neutral movements. However, against our prediction, our results suggest that nonpain motor goal pursuit did not affect pain-related somatosensory attention (hypothesis 2.b), as reflected by the lack of difference in amplitude between the threat and the conflicting components.

A graphic representation of the N120 and P200 components is showed in Fig. 2.

### Secondary outcomes

*Pain SEPs.* Paired t-test comparisons between SEPs elicited during the execution of threat and conflicting movements showed no significant differences for the N120 component ($P > 0.05$). Similarly, the P200 component did not show any significant differences in amplitude between conflicting and threat movements ($P > 0.05$). Contrary to what was initially hypothesized, this suggests that the presence of a reward did not modulate the pain-SEPs.

### Manipulation check

Results from the one-way ANOVA on the self-reported measures of fear of movement revealed a main effect of movement type, $F_{(1.168,44)} = 77.81$, $P < 0.001$, $\eta_p^2 = 0.62$. In particular, *post-hoc* tests showed that participants were more fearful to execute threat actions (M = 46.17, SD = 26.65) compared to neutral $t_{(44)} = -10.88$, $P_{bonf} < 0.001$, $d = -1.62$, M = 13.42, SD = 14.69. Similarly, executing conflicting actions (M = 43.44, SD = 26.51) evoked more fear than neutral ones $t_{(44)} = -9.97$, $P_{bonf} < 0.001$, $d = -1.49$, while no significant differences emerged between threat and conflicting actions, $t_{(44)} = 0.91$, $P_{bonf} = 1.00$, $d = 0.14$.

Similarly, the one-way ANOVA on pain expectancy, showed a main effect of movement type, $F_{(1.398,44)} = 180.870$, $P < 0.001$, $\eta_p^2 = 0.804$. *Post-hoc* tests showed that participants expected the pain stimulus significantly more in the threat $t_{(44)} = -16.15$, $P_{bonf} < 0.001$, $d = -2.41$, (M = 58.27, SD = 16.96), and conflicting movements, $t_{(44)} = -16.77$, $P_{bonf} < 0.001$, $d = -2.50$, (M = 59.88, SD = 15.03) compared to neutral movements (M = 16.42, SD = 16.18), additionally no significant differences emerged in pain expectancy between threat and conflicting action, $t_{(44)} = -0.62$, $P_{bonf} = 1.00$, $d = -0.092$. Moreover, participants did not report significantly different scores regarding the intensity of the pain, $t_{(44)} = -0.67$, $P = 0.51$, $d = 0.80$, they received while executing a conflicting (M = 61.42, SD = 14.95) or threat action (M = 61.99, SD = 14.97).

Finally, the one-way ANOVA carried out on participants free choices, showed a main effect of movement type, $F_{(1.592,42)} = 11.82$, $P < 0.001$, $\eta_p^2 = 0.22$, where participants chose the conflicting movements significantly more (M = 6.23, SD = 3.91) than the threat (M = 1.91, SD = 2.40), $t_{(42)} = -4.85$, $P_{bonf} < 0.001$, $d = -0.74$, and the neutral movements (M = 3.86, SD = 3.62) $t_{(42)} = -2.662$, $P_{bonf} = 0.028$, $d = -0.41$. Furthermore, no *post-hoc* differences emerged between the number of times participants chose a threat and neutral movement, $t_{(42)} = 2.19$, $P_{bonf} = 0.09$, $d = 0.33$. This suggests that overall participants preferred the conflicting movement, in which they could also receive a reward. An overview of all self-reports and behavioral variables are reported in Table 2, with mean, standard deviation, and range.

### Motor performance accuracy

The one-way ANOVA did not reveal significant differences in the motor accuracy in executing the three movements, $F_{(1.796,44)} = 0.53$, $P = 0.571$, $\eta_p^2 = 0.012$. Participants hit the boundaries while executing a threat movement an average of 1.87 times (SD = 0.35, range 1.17–2.96), 1.94 during a neutral movement (SD = 0.49, range 0.83–3.10) and 1.93 times during a conflicting movement (SD = 0.37, range 1.36–3.16).

## DISCUSSION

The aim of this study was to investigate whether the threat of pain and nonpain motor goal pursuit modulate pain-related somatosensory attention. More specifically, we measured SEPs from a pain-relevant and a pain-irrelevant body location during the execution of motor actions paired with threatening, conflicting and neutral consequences. We expected the threat of pain to enhance somatosensory attention. This would have been reflected by a

**Table 2 Descriptives and mean self-reports.**

|  | Mean (Range) | SD |
|---|---|---|
| Age | 22.47 (18–44) | 4.26 |
| Importance pain avoidance | 47.86 (0–100) | 31.67 |
| Importance winning lottery tickets | 64.63 (0–100) | 29.25 |
| Stimulus intensity (In mA) | 31.44 (8–68) | 14.41 |
| Self-report pain intensity | 7.67 (6–8) | 0.56 |
| Free choice conflicting | 6.23 (51.92%) | 3.91 |
| Free choice neutral | 3.86 (30.67%) | 3.62 |
| Free choice threat | 1.91 (15.92%) | 2.40 |
| Collisions conflicting | 1.93 (1.36–3.16) | 0.37 |
| Collisions neutral | 1.94 (0.83–3.10) | 0.49 |
| Collisions threat | 1.87 (1.17–2.96) | 0.35 |
| Fear of movement conflicting | 43.44 (0–93.11) | 26.51 |
| Pain expectancy conflicting | 59.88 (21.50–89.89) | 15.03 |
| Pain intensity conflicting | 61.42 (14.83–83.83) | 14.95 |
| Fear of movement neutral | 13.42 (0–54.83) | 14.69 |
| Pain expectancy neutral | 16.42 (0–71.33) | 16.18 |
| Fear of movement threat | 46.17 (0–97.56) | 26.65 |
| Pain expectancy threat | 58.27 (15.33–94.67) | 16.96 |
| Pain intensity threat | 61.99 (13.67–84) | 14.97 |

larger N120 in the pain-relevant location and a location unspecific enhancement of the P200 component for threat movements compared to neutral movements. Furthermore, we expected nonpain motor goal pursuit to inhibit pain-related somatosensory attention as reflected by a smaller N120 component in the pain-relevant location and a location unspecific reduction of the P200 component for conflicting movements compared to threat movements.

However, for what concerns the N120 component, results did not support our hypothesis. In fact, we did not find significant differences in the amplitude of this component across movements conditioned with neutral, threatening, or conflicting outcomes. This suggests that our motivational manipulation (*i.e.*, the possibility to get a reward for a good motor performance) did not affect the amplitude of the N120 component. Moreover, based on previous findings (*Clauwaert et al., 2018, 2020*), we also expected a larger N120 in response to stimuli applied on the pain-relevant body location compared to a pain-irrelevant body location (*i.e.*, spatial attention bias by pain). Yet, we did not replicate the finding on the N120. A possible explanation may be the methodological differences between the current and previous similar studies (*Clauwaert et al., 2018, 2020*). In fact, in the previous experiments, movements were performed with either the left or the right hand and the pain and the tactile stimuli could be administered to the left or the right hand. This implied that the pain-related *vs* the non-pain-related actions consisted of two different movements executed with distinct body parts.

Furthermore, the movement consisted in simple reaching actions, while in our paradigm, actions are more complex. In our current experiment, because the action required was more complex, participants always performed the movement with the right (dominant) hand but towards different directions. The tactile stimuli were administered to either the wrist or the triceps tendon of the right arm only. Possibly, and similarly to existing literature (*Torta et al., 2020*), stimulating two proximal body locations on the same limb might have weakened the modulation of the SEPs by pain. This is further supported by studies showing that the pain and the somatosensory systems follow a proximal-to-distal gradient of sensitivity and have a similar topographic organization (*Andersson et al., 1997*; *DaSilva et al., 2002*; *Strigo et al., 2003*; *Tarkka & Treede, 1993*; *Vanden Bulcke et al., 2014*; *Vogel et al., 2003*; *Weissman-Fogel, Brayer-Zwi & Defrin, 2012*).

Concerning the P200, our results showed that the amplitude of this component was modulated by the movement type, but in a different direction than what was hypothesized. In conformity with our hypothesis, the P200 was significantly larger for threat movements compared to neutral ones, regardless of the location. However, contrarily to our hypothesis, no differences emerged between threat and conflicting actions. This could be due to the fact that in the conflicting movement, pain stimuli were still occurring. Therefore, the threat of pain might have induced an increased vigilant state towards somatosensory inputs occurring during pain-related actions, increasing somatosensory attending and therefore the P200 amplitude. Moreover, and in line with our hypothesis, we did not find an effect of stimulus location on this component, which is in line with previous studies that found location unspecific differences in amplitude and reflects a general state of arousal (*Clauwaert et al., 2018*, *2020*).

Regarding the results on the SEPs elicited by pain stimuli, results did not show significant differences in the SEPs amplitude in the threat *vs* conflicting movements. As in the case of tactile SEPs, a possible interpretation of this effect could be that the expectation of pain stimuli facilitated top-down facilitation of somatosensory inputs, fostering stronger somatosensory attending responses. Furthermore, the fact that we observed a difference in the P200 evoked by the tactile stimuli but not in the P200 evoked by the pain stimuli could be explained by the number of trials that we averaged to obtain these components. In fact, the tactile P200 was obtained—for each movement type and stimulus location—by averaging 36 trials. On the other hand, the pain evoked P200 was exported by averaging 12 trials per movement type. Hence, the reduced number of trials in the pain SEPs might have weakened this effect. However, with stimuli of high intensity like the ones implemented in this task, there is a better signal-to-noise ratio which might have helped containing the impact of the low number of trials for the pain-SEPs.

Concerning the behavioral results, we can conclude that the conditioning procedure was successful given that participants reported higher ratings of fear of movement and pain expectancy for the threat and conflicting movements compared to the neutral ones.

Intuitively, one might expect that the reward would affect not only attention but also fear and pain intensity. However, participants reported similar pain intensity and fear of movement for threat and conflicting actions suggesting that the reward did not affect the perception of pain or pain-related fear. However, as reported in our pre-registration

(osf.io/vbgru), no specific *a priori* hypothesis on the inhibitory role of reward on fear and pain intensity was formulated since previous studies obtained inconsistent results (*Claes et al., 2014*; *Meulders et al., 2015*). Furthermore, these measures were added to the protocol to check whether pain increased fear, pain intensity and expectancy, rather than to test the impact of reward on fear and pain intensity. In fact, measures of fear and pain intensity were obtained at the end of each block as a manipulation check, rather than trial-by-trial. To obtain a better understanding of the role of reward on fear and pain intensity, future studies should be designed with the specific objective of tackling these constructs.

In general, perhaps, our attempt of manipulating the motivational value of executing the action (reward manipulation) might have been not strong enough to activate goal-shielding processes and, consequently, the elaboration of pain-related information might have been prioritized over motor goal-related information. This would be indicative of stronger bottom-up attentional capturing mechanisms by pain, rather than inhibition of attentional responses towards pain. In line with this, it is known that despite bottom-up and top-down attention are different processes, they share the same neural system (*i.e.*, the frontoparietal network), and influence each other to orient attention (*Katsuki & Constantinidis, 2014*). Hence, our results might reflect a synergic bottom-up and top-down attentional processing. Nonetheless, a limitation of this study is that we are unable to directly assess the processing of goal-related information from the current paradigm. Despite this, participants reported to value "obtaining the reward" quite high (suggesting a strong engagement with the task) and they tended to select the conflicting movement more often during the free choice phase, showing motivation in choosing this movement despite the occurrence of pain. These results seem to suggest that they were quite motivated in executing the task. However, because we did not include direct measures of goal-shielding processes we cannot conclude whether this motivation was also responsible of inhibition of pain-related information. Therefore, future studies should address this facet in order to be able to conclude that pursuing nonpain motor goal pursuit attenuates pain-related somatosensory attention.

Another limitation of the current study is that examining SEPs might not be an exclusive indicator of attention. However, this study was designed in line with previous findings that used similar methodologies and interpreted the results in terms of attentional mechanisms (*Clauwaert et al., 2018*, *2020*). Moreover, we targeted these SEPs that, in previous EEG studies, have shown to be susceptible to attentional processes (*Eimer & Forster, 2003*; *Fiorio et al., 2012*; *Franz et al., 2015*; *García-Larrea, Bastuji & Mauguière, 1991*; *García-Larrea, Lukaszewicz & Mauguière, 1995*; *Zopf et al., 2004*).

## CONCLUSIONS

This study demonstrates that executing actions conditioned with painful outcomes enhances somatosensory processing and this could be driven by top-down facilitation of somatosensory attending due to the anticipation of painful events. Crucially, we did not observe attentional modulation by nonpain motor goals. This study provides new insights into the interaction between goals and pain, with a special focus on the context of motor control which is often strongly related to pain. This work highlights the importance of

conducting more research on how fostering non-pain motor goal pursuit can facilitate the processing of goal-related information over the threat of pain. Implementing these insights into the development of personalized clinical interventions could potentiate the analgesic effects of the motor system on pain, leading to profound implications for pain management.

## ACKNOWLEDGEMENTS

The authors wish to thank Jacco Ronner and Erik Bongaerts for programming the experiment, Richard Benning for graphical support of the experiment, Behnam Z. Mortazavi and Nicolas Deuster for their assistance in data collection and Pascal Mestdagh for the technical support. The authors report no conflict of interest.

### Funding

This research was conducted at Maastricht University and was supported by a Research Grant from the Research Foundation-Flanders (FWO), Belgium, (grant ID G000518N) granted to Stefaan Van Damme, Diana Torta and Ann Meulders. The contribution of Ann Meulders was supported by a Vidi grant from the Netherlands Organization for Scientific Research (NWO), The Netherlands (grant ID 452-17-002). The funders had no role in study design, data collection and analysis, decision to publish, or preparation of the manuscript.

### Grant Disclosures

The following grant information was disclosed by the authors:
Maastricht University.
Research Grant from the Research Foundation-Flanders (FWO), Belgium: G000518N.
Vidi Grant from the Netherlands Organization for Scientific Research (NWO), The Netherlands: 452-17-002.

### Competing Interests

The authors declare that they have no competing interests.

### Author Contributions

- Eleana A. Pinto conceived and designed the experiments, performed the experiments, analyzed the data, prepared figures and/or tables, authored or reviewed drafts of the article, and approved the final draft.
- Stefaan Van Damme conceived and designed the experiments, authored or reviewed drafts of the article, and approved the final draft.
- Diana M. Torta conceived and designed the experiments, authored or reviewed drafts of the article, and approved the final draft.
- Ann Meulders conceived and designed the experiments, authored or reviewed drafts of the article, and approved the final draft.

## Human Ethics

The following information was supplied relating to ethical approvals (*i.e.*, approving body and any reference numbers):

The study was conducted in accordance with the Ethics Review Committee Psychology and Neuroscience (ERCPN) of Maastricht University who approved the experimental protocol (number ERCPN- 233_18_02_2021_A1).

## Data Availability

The raw data are available in the Supplemental File.

## Supplemental Information

Supplemental information for this article can be found online at http://dx.doi.org/10.7717/peerj.16544#supplemental-information.

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
