# Peer review of "Modulation of attention to pain by goal-directed action: a somatosensory evoked potentials approach"

_PeerJ, doi:10.7717/peerj.16544_

## Round 0.1 · original submission · Minor Revisions

This revised submission of a previously rejected manuscript is greatly improved and reviewers are generally positive about this study presentation. However, there are several arguably major concerns (see Rev.2) about the mismatch of hypotheses tested. Please, clarify if this was a typo or indeed a mismatch in the study design, which would constitute a concern in the study design validity. The requested justification may be directly related to the resolution of this question.

·

Basic reporting

Review
Original title:
Modulation of attention to pain by goal-directed action

Dear Authors
I appreciate Yours efforts and commitment in researching this interesting topic. I really see how much work did You put in enchancing quality of this manuscript. I have read the response letter to each reviewer and I am now sure that the changes made by You to the manuscript already raised the quality. I will now go thoroughly thru the paper and if I will be able to – I will give my suggestions on how to perfect it.
Lenguage is professional and flow is good. the references list is now of good quality with no doubt comapring to earlier submission.

I suggest changing title to: Modulation of attention to pain by goal-directed action assessed by analysys of Somatosensory Evoked Potentials (in threat, conflict or neutral movement effects’ conditions.)

Arguemntation:
You used only SEPs for assessment which are a way to measure but not a one single present method. You checked namely three conditions. I leave the part marked red put in brackets for consideration of the Authors and the Editor (in my opinion it could be added but not crucial).

Abstract
The text cited below:
We expected that threat of pain would enhance somatosensory attention, as reûected by a larger N120 in the pain-relevant location and a location unspeciûc enhancement of the P200 component for threat movements compared to neutral movements. Furthermore, we expected that the possibility to get a reward would inhibit pain-related somatosensory attention as reûected by a smaller N120 component in the pain-relevant location and a location unspeciûc reduction of the P200 component for conûicting movements compared to threatening movements.
Should be placed in the end of introduction. There are other options: either delete it, or rewrite it in a way to show only methods, not hypotheses.

Figure 1 – please ensure the best possible quality – in the file delivered for review the quality of image is poor
Line 70 – remove ‘the’ after ‘impact’
Lines 80-81 – not adequate example – you are talking about decision making about undertaking some action which potentially may bring pain, which is something other than managing or controlling movement in a way to avoid pain and something other than having to move despite felling pain. Please give appropriate example.
Line 108 – it may be surprising to You dear Authors, but with all the respect - that is only your opinion. Please, bring scientific and informative sentences to the readers and avoid subjectification.

Experimental design

Exclusion criteria – was taking drugs, e.g – painkillers an exclusion criteria or not? Why there is now word about it?
Another important thing in pain studies including woman is an information if femle participants were in the period time, or any other stage of the cycle. (Choi, J. C., Park, S. K., Kim, Y. H., Shin, Y. W., Kwon, J. S., Kim, J. S., ... & Lee, M. S. (2006). Different brain activation patterns to pain and pain-related unpleasantness during the menstrual cycle. The Journal of the American Society of Anesthesiologists, 105(1), 120-127.)
The last one I would like to emphasize in Times when depression is a common problem – the exclusion criteria should also include mental problems – see (Pérez‐Aranda, A., Hofmann, J., Feliu‐Soler, A., Ramírez‐Maestre, C., Andrés‐Rodríguez, L., Ruch, W., & Luciano, J. V. (2019). Laughing away the pain: A narrative review of humour, sense of humour and pain. European Journal of Pain, 23(2), 220-233.) and (Linton, S. J., & Bergbom, S. (2011). Understanding the link between depression and pain. Scandinavian Journal of Pain, 2(2), 47-54.)

Validity of the findings

The results and its presentation are valid and informative.

Additional comments

Since Your study design is interesting, please consider puttin a picture of a participant (consent needed) while performing the task – showing the setup including some detailed information and putting it around line 180. Fig 1 which you already have is very small – it does not show anything in detail
Line 314 – space between value and Hz ;)

Wishing You all the best with Your future research!
Jarosław Muracki, PhD

·

Basic reporting

Major points:
1. The authors tested pain-related somatosensory attention by assessing N120 and P220 as the neural signature of such process. And specific hypotheses are built upon it. However, the mechanistic interpretation of these two components underlying this assumption is missing. This should be discussed before introducing specific hypothesis in the introduction section. (Line 113-124) For example, what’s the mechanistic implication of N120 and P220?


2. The hypothesis described in the introduction (Line 117-124) and Data analyses (Line 346-363) are mismatched as shown below. Mismatched hypotheses are marked in red. I encourage the author to go through the rest of the paper since mislabeling occurred elsewhere.

In the Introduction section,
This would be reflected by 1.a) a larger N120 in the pain-relevant location in the threat movements as compared to neutral movements and 1.b) a location unspecific enhancement of the P200 component for threat movements compared to neutral movements. Furthermore, we expected the nonpain motor goal pursuit to inhibit pain- related somatosensory attention as reflected by 2.a) a smaller N120 component in the pain-relevant location in the conflicting movements as compared to threat movements and 2.b) a location unspecific reduction of the P200 component for conflicting movements compared to threat movements.
In the Data analysis section,
To test whether the threat of pain (hypothesis 1.a) or nonpain motor goal pursuit (hypothesis 1.b) modulates pain-related somatosensory attention (operationalized as the amplitude of the N120 component elicited by the tactile stimuli at the pain-relevant location),
​​Similarly, to test whether the threat of pain (hypothesis 2.a) or nonpain motor goal pursuit (hypothesis 2.b) modulates pain-related somatosensory attention (operationalized as the amplitude of the P200 component elicited by the tactile stimuli regardless of the location)

3. The author found insignificant modulation of movement types on N120 while significant modulation on P200. The possible physiological interpretation is missing.

Minor points:
1. The items inside the computer monitor are blurred in the figure 1. I suggest showing the screenshot of each task epoch as it's alone. Appropriate labeling of each item on the screen could be helpful for explaining the task procedure.
2. Table 1: It should be threat condition instead of Negative condition
3. Line 108: the word ‘surprising’ is too emotional.

Experimental design

The author needs to justify the basis of identifying SEP components only based on a specific set of electrodes (N120 on Fz, Fc1, Fc2 and Cz; P200 on Cz Fc1, Fc2, Cp2 and Cp1), either through literature reference or statistical analyses. Moreover, the procedure of the ROI selection was not mentioned.
Line 334-339

Validity of the findings

no comment

---

## Round 0.2 · accepted · Accept

Thank you for your careful revision.

·

Basic reporting

The authors have made significant improvements to the manuscript. All major concerns have been addressed by the corrections.

Experimental design

no comment

Validity of the findings

no comment